# Single-Step Fabrication of Longtail Glasswing Butterfly-Inspired Omnidirectional Antireflective Structures

**DOI:** 10.3390/nano12111856

**Published:** 2022-05-29

**Authors:** Chung-Jui Lai, Hui-Ping Tsai, Ju-Yu Chen, Mei-Xuan Wu, You-Jie Chen, Kun-Yi Lin, Hong-Ta Yang

**Affiliations:** 1Department of Chemical Engineering, National Chung Hsing University, 145 Xingda Road, Taichung City 40227, Taiwan; martin50312@yahoo.com.tw (C.-J.L.); page52026@gmail.com (J.-Y.C.); meixuan316@gmail.com (M.-X.W.); jerry3771740@gmail.com (Y.-J.C.); 2Department of Civil Engineering, National Chung Hsing University, 145 Xingda Road, Taichung City 40227, Taiwan; huiping.tsai@nchu.edu.tw; 3Department of Environmental Engineering, National Chung Hsing University, 145 Xingda Road, Taichung City 40227, Taiwan

**Keywords:** antireflective nanostructures, longtail glasswing butterfly, subwavelength colloids, single step, antireflection

## Abstract

Most bio-inspired antireflective nanostructures are extremely vulnerable and suffer from complicated lithography-based fabrication procedures. To address the issues, we report a scalable and simple non-lithography-based approach to engineer robust antireflective structures, inspired by the longtail glasswing butterfly, in a single step. The resulting two-dimensional randomly arranged 80/130/180 nm silica colloids, partially embedded in a polymeric matrix, generate a gradual refractive index transition at the air/substrate interface to suppress light reflection. Importantly, the randomly arranged subwavelength silica colloids display even better antireflection performance for large incident angles than that of two-dimensional non-close-packed silica colloidal crystals. The biomimetic coating is of considerable technological importance in numerous practical applications.

## 1. Introduction

Fresnel reflection occurs when light penetrates through an interface between two optical media with different refractive indices [1,2]. The light reflection from optical systems, such as a camera lens, corrective lenses, panel displays, automotive dashboards, and face shields incurs veil glare and degrades overall optical performance. To diminish the unfavorable light reflection, single-layer/multilayer interference coatings consisting of transparent materials with appropriate refractive indices have been widely used to moderate the refractive index change [3,4,5,6]. However, low-refractive-index transparent materials are scarce and highly priced. As an alternative, porous coating layers with adjustable refractive indices can be fabricated through phase separation of polymer blends, sol-gel processing, plasma-enhanced chemical vapor deposition, multilayer deposition of nanoparticles/polyelectrolytes, and many others [7,8,9,10,11]. Unfortunately, the porous antireflection coatings are limited by narrow operating wavelength regions.

Over four hundred million years of natural selection, living beings have created diverse functional architectures for survival. For example, moth eyes, hawk moth wings, and cicada wings are covered with hexagonally arranged submicrometer-scale protuberances to reduce light scattering as well as to minimize reflectivity in the broadband wavelength region over wide incident angles [12,13,14]. The high-aspect-ratio structure arrays allow the insects to avoid tracking by predators. Inspired by their camouflage characteristics, artificial conical structure arrays, pillar-shaped structure arrays, pyramid-like structure arrays, and so on, have been developed via a large variety of top-down fabrication approaches to function as antireflective structures [15,16,17]. Nevertheless, current lithography-based technologies, such as interference lithography and photolithography, are costly, complex, and restricted to either a limited sample size or low resolution features.

By contrast, spontaneous crystallization of colloids renders an inexpensive and simple alternative for creating high-resolution antireflective structures. The self-assembled colloidal monolayers can serve as templates to pattern demanded architectures [18,19]. However, most of existing bottom-up fabrication approaches, including breath figure-based assemblies, magnetic-/electro-filed-assisted assemblies, capillary-force-induced assemblies, and Langmuir-Blodgett technology inevitably suffer from low-throughput production and technical incompatibility with standard industrial microfabrication [20,21,22,23,24]. In addition, the self-assembled colloids thermodynamically favor hexagonal close-packed crystal structures, while non-close-packed antireflective structures are required to clamp down the internal reflection between structures. To resolve the issues, a spinning-shear-force-induced assembly technology has recently been developed to rapidly produce wafer-sized non-close-packed colloidal monolayers using a standard spin-coater [25]. The shear-aligned colloidal crystals can be utilized to design and build moderate-aspect-ratio (~1) antireflective structures [26,27]. Unfortunately, their antireflection performances are impaired at large viewing angles.

Longtail glasswing butterfly wings are covered with irregular positioning moderate-aspect-ratio dome-shaped structures to produce angle-independent optical transparency [28,29]. Interestingly, the randomness is not only found on the structure arrangement, but also on the structural height distribution. The dome-shaped structures feature a Gaussian height distribution and possess an average inter-structure distance of ~100 nm. By optimizing the structure height and distribution, the large-scale fabrication of such antireflective structures seems feasible. Taking the longtail glasswing butterfly as a prototype, varisized submicrometer-scale colloids are spin-coated to biomimic the irregular dome-shaped structures in this research. The corresponding omnidirectional antireflective characteristics are evaluated to bridge the bioinspired structures and practical applications.

## 2. Materials and Methods

Spherical silica colloids with less than 5% diameter variation are synthesized by following the well-established StÖber method [30]. In a standard synthesis, the amount of tetraethyl orthosilicate (TEOS) (≥99 wt.%, Merck & Company, Incorporated) demanded is rapidly poured into a mixture of 650 mL of anhydrous ethanol (200-proof, Merck & Company, Incorporated, Kenilworth, NJ, USA), 55 mL of deionized water, and 25 mL of ammonium hydroxide solution (28 wt.%, Merck & Company, Incorporated). The solution is stirred vigorously under ambient conditions for 24 h to bring about monodispersed silica colloids. The colloid diameter can be easily adjusted through adding varied amounts of TEOS in the solution, where 29, 34, and 42 mL of TEOS are applied to synthesize silica colloids with diameters of 80 nm, 130 nm, and 180 nm, respectively, in this study. The as-synthesized silica colloids are then purified in anhydrous ethanol by repeating five centrifugation−ultrasonication processes to remove any impurity.

Spherical StÖber silica colloids with tunable diameters are dispersed in photocurable ethoxylated trimethylolpropane triacrylate (ETPTA) (Sartomer Company Corporation, Messe Düsseldorf, Düsseldorf, Germany) monomers with a colloid volume fraction of 25%. 2-hydroxy-2-methyl-1-phenyl-1-propanone (Darocur 1173, BASF Corporation, Paris, France) is added as a photoinitiator. The viscosity of the resulting suspension is determined by the colloidal volume fraction. A higher colloidal volume fraction leads to a more viscous suspension. The as-prepared suspension is deposited onto an RCA-cleaned poly (ethylene terephthalate) (PET) (Ensinger Incorporation, Hsinchu, Taiwan) substrate, followed by a spin-coating process (5000 rpm for 2 min) to uniformly spread the silica colloids. The polymerization of ETPTA monomers can be greatly increased by employing Darocur 1173 in a UV curing chamber (XLite 500, OPAS Corporation, Taichung, Taiwan).

## 3. Results and Discussion

To investigate the dependences of silica colloid size and arrangement on antireflection characteristics, non-close-packed 80 nm silica colloidal crystals, non-close-packed 130 nm silica colloidal crystals, and non-close-packed 180 nm silica colloidal crystals are spin-coated on PET substrates, respectively (Figure 1). Instead of irregularly positioning, long-range hexagonal orderings of the self-assembled silica colloids are clearly found. The colloids are with a fixed inter-colloid distance of $\sqrt{2}$ D, where D represents the silica colloid size. Even if a few defects are observed, they do not significantly affect the optical uniformity of the silica colloidal crystal-coated substrates. The observed colloidal crystallization is attributed to shear-induced ordering. Obviously, the optical transparencies of the substrates are improved with increasing of the silica colloid size. It is believed that larger colloids can establish smoother refractive index gradients on the surfaces, and therefore reduce the optical reflection effectively. Unanticipatedly, the PET substrate turns blue after coating with 180 nm silica colloidal crystals. The blue color is derived from considerable light scattering, which is more effective at short wavelengths, in all directions by the 180 nm silica protuberances [31].

To address the issue, the spin-coating technology is utilized to engineer two-dimensional randomly arranged subwavelength silica colloids. Spherical silica colloids with diameters of 80 nm, 130 nm, and 180 nm, synthesized by a standard StÖber method, are dispersed in photocurable ETPTA monomers with a colloid volume fraction of 25%. Bioinspired by the longtail glasswing butterfly wings, the amounts of 80 nm colloids, 130 nm colloids, and 180 nm colloids in the silica colloidal suspension are adjusted in a ratio of 1:2:1. The as-prepared silica colloidal suspension is deposited and spread on a PET substrate in a spin-coating process. The ETPTA monomers are finally UV-polymerized to bring about a monolayered silica colloid/poly(ETPTA) composite.

As displayed in Figure 2, the irregularly positioning silica colloids are about 100 nm apart from neighboring colloids, and partially embedded in the poly(ETPTA) matrix. The silica protuberances are able to moderate the abrupt refractive index change at the air/PET substrate interface to suppress optical reflection. It is observed that a bare PET substrate is milky-white in appearance under white light illumination (Figure 3a). In comparison with that, a considerable improvement in optical transparency from the randomly arranged 80/130/180 nm silica colloid-coated PET substrate is clearly evident. Although the substrate is slightly blue, primarily caused by the Rayleigh scattering from 180 nm silica colloids, the high uniformity of the antireflection coating is apparent from the photo [32]. To investigate the antireflection functionalities of the coating, normal-incidence specular optical reflection and transmission measurements of the PET substrates are carried out using a UV-Visible-NIR spectrometer (HR4000, Ocean Optics Incorporation, Shanghai, China). It is found that the bare substrate displays a reflectance of ~7% in the visible spectrum from 350 to 800 nm, matching well with the estimated value using the Fresnel equation [33] Importantly, the reflectance of the randomly arranged silica colloid-coated substrate is reduced by ~4% over the whole visible wavelength region. Besides, this silica colloid-coated substrate exhibits consistently higher transmittance than that of the bare substrate. By contrast, a single poly(ETPTA) layer-coated substrate and the bare substrate present similar reflectances and transmittances (Figure 3b). The results disclose that the anti-glare capability can be enhanced by introducing the longtail glasswing butterfly-inspired structures, but not the single polymer layer.

To further evaluate their corresponding omnidirectional antireflection characteristics, optical reflection and transmission measurements of all the above-mentioned PET substrates are acquired in the visible spectrum at varied incident angles. The measurements from five different regions of each substrate are averaged and compared in Figure 4a,b. It is noticed that the average reflectance of an uncoated PET substrate (black solid curve) is increased from 7% to 34%, while its average transmittance is decreased from 90% to 64% as the incident angle varies from 0° to 75°. Compared with that, Fresnel reflections at varied incident angles are reduced by spin-coating non-close-packed silica colloidal crystals on the substrates. Larger silica colloids are able to generate more gradual refractive index transitions even for large incident angles, leading to lower average reflectances and higher average transmittances for whole the angles of incidence. However, despite the fact that 180 nm silica colloidal crystal-coated substrate (blue dashed curve) displays the lowest average reflectances, the resulting light scattering diminishes the optical transmission of the substrate. In contrast, the 80/130/180 nm silica colloid-coated substrate (red solid curve) displays similar average reflectance with these silica colloidal crystal-coated substrates at 0°. Importantly, its average reflectances are getting even lower than the others for larger incident angles as a result of the gradual refractive index change. Owing to the inconsiderable light scattering by the sparse 180 nm silica protuberances, similar evolution trends are found on the average transmittance curves. The average transmittance of the PET substrate can be increased by 4% at 0°, and even by 11% at 75°. Clearly, in comparison with the bare substrate, the randomly arranged silica colloid-coated substrate remains transparent at a large viewing angle of 75° (Figure 4b). The photos further confirm that the randomly arranged 80/130/180 nm silica colloids can be applied to improve omnidirectional antireflective properties.

The antireflective properties are further complemented by plotting calculated effective refractive index ($n_{eff}$) changes of the hemispherical silica protuberances from the polymer surfaces to the tops of the protuberances. On the basis of effective medium theory, $n_{eff}$ at any protuberant height (*h*) is expressed as $n_{eff}=\sqrt{n_{silica}^{2}\times f_{silica}+n_{air}^{2}\times f_{air}}$, where $n_{silica}=1.42$, $n_{air}=1$, $f_{silica}$ and $f_{air}$ represent the fractions of silica and air, respectively. For the hexagonally non-close-packed silica colloidal crystals, the average inter-colloid distance is evident to be 2√2 R, where R denotes the radius of silica colloids (40, 65, and 90 nm in this study), indicating that $f_{silica}=\frac{\left(R^{2}-h^{2}\right)\pi }{4\sqrt{3}R^{2}}$, $f_{air}=1-\frac{\left(R^{2}-h^{2}\right)\pi }{4\sqrt{3}R^{2}}$. This leads to the formula $n_{eff}=\sqrt{\frac{\left(n_{silica}^{2}-n_{air}^{2}\right)\left(R^{2}-h^{2}\right)\pi }{4\sqrt{3}R^{2}}+n_{air}^{2}}=\sqrt{\frac{n_{silica}^{2}\left(R^{2}-h^{2}\right)\pi }{4\sqrt{3}R^{2}}+\frac{n_{air}^{2}4\sqrt{3}R^{2}-n_{air}^{2}\left(R^{2}-h^{2}\right)\pi }{4\sqrt{3}R^{2}}}$ By approximating the average inter-colloid distance of the 80/130/180 nm silica colloid-coated substrate also equals to $2\sqrt{2}R$, $n_{eff}$ of the coating can be expressed as $n_{eff\left(\mathrm{mix}\right)}=\frac{1}{4}\times n_{eff\left(R=40\mathrm{nm}\right)}+\frac{1}{2}\times n_{eff\left(R=65\mathrm{nm}\right)}+\frac{1}{4}\times n_{eff\left(R=90\mathrm{nm}\right)}$. Figure 5 compares the calculated $n_{eff}$ profiles of the as-prepared silica colloid-coated substrates. For the 80/130/180 nm silica colloid-coated substrate (red dashed curve), the $n_{eff}$ gradually changes from 1.42 (at the bottom of protuberance) to 1 (air), resulting in the lowest reflection in the whole visible spectrum.

Coating hardness, the capacity of a particular coating to withstand superficial mechanical forces, is an important factor to assess its environmental durability. In this work, a constant-load scratch test according to the ASTM D3363 method is applied to determine the hardness of as-coated randomly arranged 80/130/180 nm silica colloid/poly(ETPTA) composite using a pencil set with different hardness grades (1H–5H) [34,35,36,37]. In the experiment, the selected pencil with a load of 10 N is placed on the coating and forms a 45° angle with the surface. The highest pencil grade that is unable to cause damage after 500 rubbing cycles is considered as the coating hardness. As displayed in Figure 6, the average reflectances and transmittances of the randomly arranged 80/130/180 nm silica colloid-coated PET substrate are well-maintained after a 4H pencil hardness test. In contrast, its intrinsic antireflection characteristics are impaired after scratching with a 5 H pencil, resulting from obvious traces of scratches on the surface (insert of Figure 6). The coating hardness is even competitive with commercial antireflection coatings [38]. Moreover, in comparison with the hardness of the randomly arranged 80/130/180 nm silica colloid-coated PET substrate, it is evident that the surface and the corresponding optical properties of a bare PET substrate are damaged after scratching with a 2H pencil (Figure 7). It is believed that the incorporation of silica colloids as reinforcing fillers improves the coating hardness, which is desirable for practical applications.

## 4. Conclusions

To conclude, a scalable approach is developed to engineer longtail glasswing butterfly wing-inspired antireflective structures in a single-step. The resulting subwavelength-scale silica colloid/polymer composite behaves with broadband antireflection characteristics and a competitive coating hardness. Importantly, in comparison with non-close-packed silica colloidal crystals, the randomly arranged silica colloids exhibit improved antireflection performance for large incident angles. It is worth mentioning that randomly arranged silica colloids can be easily fabricated by the scalable and microfabrication-compatible spin-coating technology. Furthermore, commercial silica colloids with large diameter variations are much more inexpensive. This low-cost and simple methodology offers a new opportunity in practical applications. Further structural optimization is still under examination and will be reported on in the near future.

## Figures and Tables

**Figure 1 nanomaterials-12-01856-f001:**
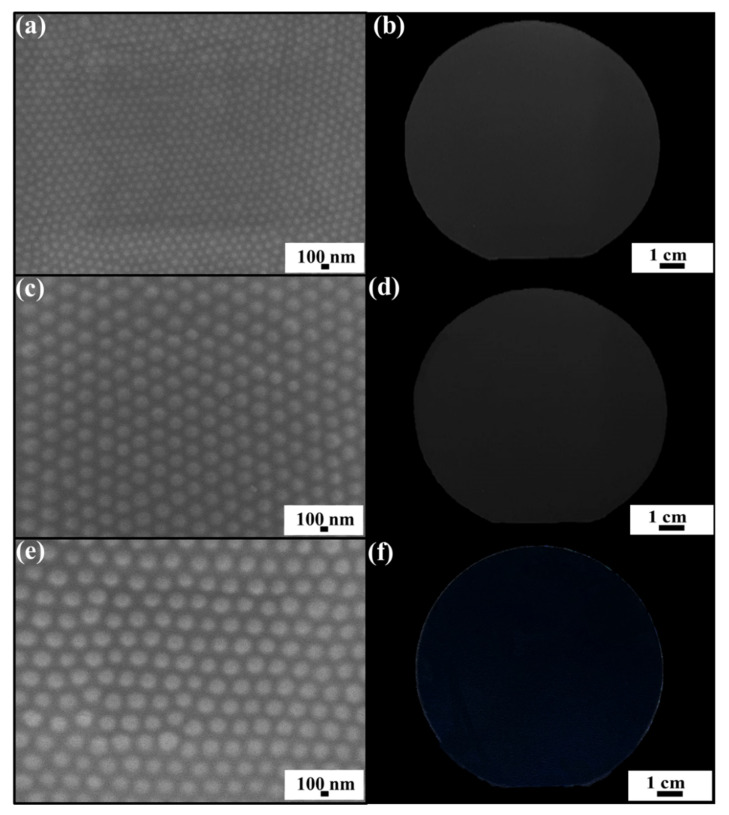
Top-view SEM images and photos of PET substrates coated with non-close-packed (**a**,**b**) 80 nm silica colloidal crystals, (**c**,**d**) 130 nm silica colloidal crystals, and (**e**,**f**) 180 nm silica colloidal crystals.

**Figure 2 nanomaterials-12-01856-f002:**
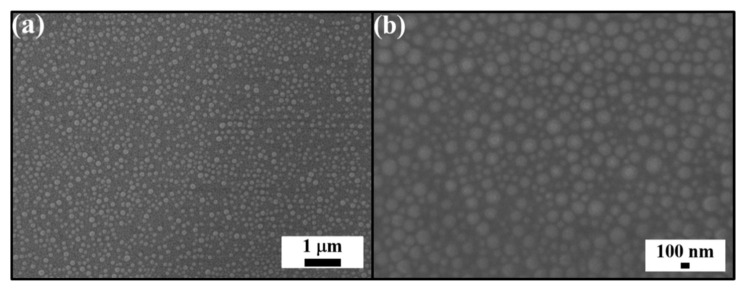
(**a**) Top-view SEM image of a randomly arranged 80/130/180 nm silica colloid-coated PET substrate. (**b**) Magnified SEM image of (**a**).

**Figure 3 nanomaterials-12-01856-f003:**
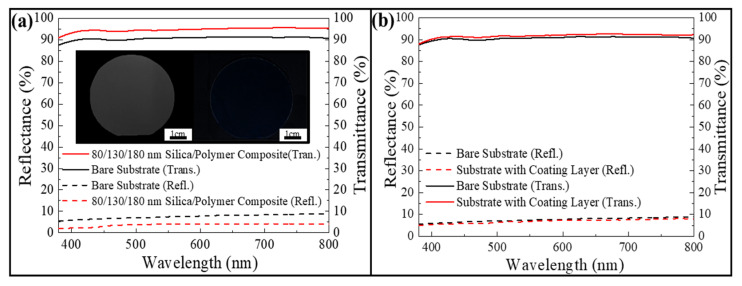
(**a**) Optical reflection spectra and optical transmission spectra in the visible wavelength region acquired from the PET substrates. Inserts showing photos of a bare PET substrate (left) and a randomly arranged 80/130/180 nm silica colloid-coated PET substrate (right). (**b**) Optical reflection spectra and optical transmission spectra in the visible wavelength region acquired from a bare PET substrate and a poly(ETPTA) layer-coated PET substrate.

**Figure 4 nanomaterials-12-01856-f004:**
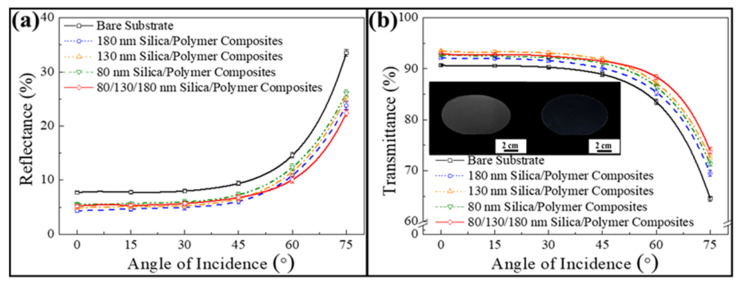
(**a**) Average reflectances and (**b**) average transmittances in the visible spectrum acquired from a bare PET substrate and silica colloid-coated PET substrates at varied angles of incidence. Inserts showing photos of a bare PET substrate and a randomly arranged 80/130/180 nm silica colloid-coated PET substrate taken from 75°.

**Figure 5 nanomaterials-12-01856-f005:**
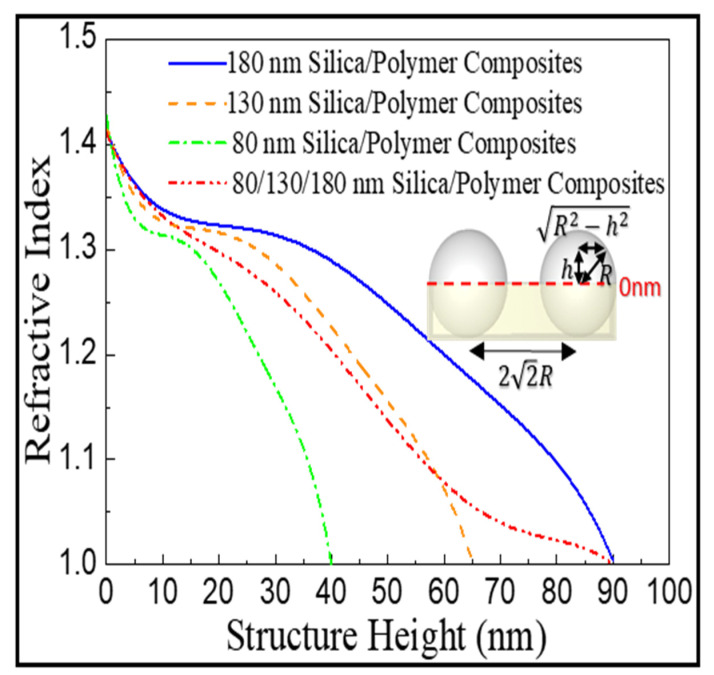
Calculated effective refractive index changes of the silica colloid-coated substrates from the polymer surface (height = 0 nm) to the top of the silica colloids.

**Figure 6 nanomaterials-12-01856-f006:**
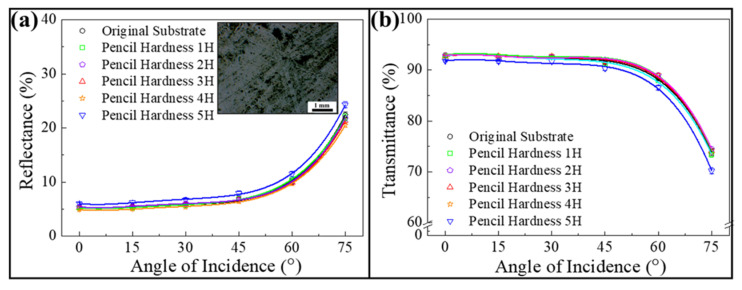
(**a**) Average reflectances and (**b**) average transmittances in the visible spectrum acquired from randomly arranged 80/130/180 nm silica colloid-coated PET substrates after pencil hardness tests at varied angles of incidence. Insert showing an optical micrograph of the randomly arranged 80/130/180 nm silica colloid-coated PET substrate after scratching with a 5H pencil.

**Figure 7 nanomaterials-12-01856-f007:**
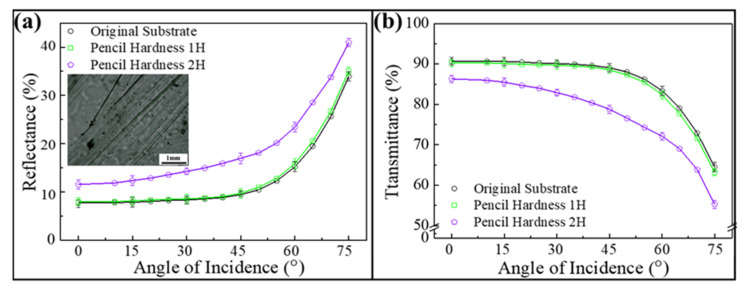
(**a**) Average reflectances and (**b**) average transmittances in the visible spectrum acquired from bare PET substrates after pencil hardness tests at varied angles of incidence. Insert showing an optical micrograph of a bare PET substrate after scratching with a 2H pencil.

## Data Availability

Data available in a publicly accessible repository.

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
