# Peer review of "Single-Step Fabrication of Longtail Glasswing Butterfly-Inspired Omnidirectional Antireflective Structures"

_nanomaterials, 2022, doi:10.3390/nano12111856_

Round 1

Reviewer 1 Report

The article is very concise, but it has some important shortcomings. The most important shortcoming concerns the particularities of the Stober reaction in order to obtain each particle size. Instead of appearing in the experimental part, we find a single reference to the synthesis of silica nanoparticles (page 3, line 104-105).
The question now is how was the particle size controlled?
How was the polydispersity? How was this controlled?
What parameters were varied to obtain colloidal systems?
There are other additions to the dispersion, and what are they?
How did you adjusted the viscosity of dispersed systems before spin-coating?
All of these questions need to be answered in this article before it will be ready for publication.

Reviewer 2 Report

Article Ref.

nanomaterials-1725106

Title:

Single-Step Fabrication of Longtail Glasswing Butterfly-Inspired Omnidirectional Antireflective Structures

Comments to the authors:

This work demonstrates a practical and relatively scalable method to produce bioinspired antireflective coatings based on the incorporation of colloidal nanoparticles of varied diameters to the film surface. The article is well organized and, in spite of being a topic which has been previously explored by several groups, the authors provide new insights concerning the combination of colloidal nanoparticles of varied diameters to enhance the omnidirectional antireflective capability of the coating.

From this reviewer point of view some issues should be addressed prior to acceptance for publication in Nanomaterials:

First of all, it is not fully clear what is the advantage of using the randomly arranged silica colloids with different diameters. The improvement in transmittance compared to the 130nm silica-polymer nanocomposite only occurs for angles higher 45º (Figure 4b), which are usually beyond those used in solar panels, for instance.

Regarding this Figure 4, the authors should explain why, when the reader compares the spectra corresponding to the 180 nm silica/polymer nanocomposite with the 80/130/180 silica/polymer nanocomposite, the transmittance of the 80/130/180 remains higher along the whole angular range while the reflectance values are exchanged above 45º. If the transmittance is higher along the whole range, the reflectance should be lower in the same range, right?

The photos of the PET coated substrates coated are not appropriate to demonstrate the antireflective effect. Usually, this is demonstrated by using brighter images in which the reflections of a treated and an untreated samples are compared. The dark images shown in this work does not provide for valuable information. For instance, inset in Figure 4 shows pictures of a bare PET substrate and silica-colloid-coated PET. The transmittance difference is ca. 10% but one is black and the other white. This is quite confusing from the reader point of view.

The references should be checked. Reference 16 seems to deal with the manufacture of pyramidal structures but, actually, not employed as anti-reflective coatings. The review by Natarajan Shanmugam et al. “Anti-Reflective Coating Materials: A Holistic Review from PV Perspective” (Energies 2020, 13, 263) includes references of AR structures based on pyramidal shapes that would fit better in this case. Also in Anti-reflective coatings: A critical, in-depth review (Energy Environ. Sci., 2011, 4, 3779).

The interrelation stablished by the authors between the scratch tests and the optical properties is not clear. To show the mechanical resistance improvement based on a pencil test, the authors should show optical micrographs or SEM images displaying the traces of the scratches. This test will leave a single scratch on the surface and, depending on the area used for transmittance measurements, this could suppose a very limited effect on the optical performance. If the authors want to stablish an interrelation between abrasion resistance and optical performance this reviewer thinks the approach employed by Fangting Chi et al. (Mechanically robust and self-cleaning antireflection coatings from nanoscale binding of hydrophobic silica nanoparticles, Solar Energy Materials and Solar Cells 200 (2019) 109939), to study a similar AR coating, is more reliable, since a felt pad would affect a larger area providing better statistical information regarding the optical response.

In addition, since the authors claim that “the incorporation of silica colloids as reinforcing fillers improves the coating hardness” there must be a demonstration of this fact by comparing the response of the samples with and without fillers. 

Round 2

Reviewer 1 Report

The article can be published in its current form.

Reviewer 2 Report

I consider that the authors have addressed most of my comments and the paper can be published as it is.

Thank you very much